# Prevention of adolescent stimulant drug use: Do the home life environment and extracurricular activities influence this? Findings from the Irish Planet Youth Survey

Fionn P. Daly[1,2], Seán R. Millar[2,3], Emmet Major[4], Peter M. Barrett[2,5,6]*

1 School of Medicine and Health, University College Cork, Cork, Ireland, 2 School of Public Health, University College Cork, Cork, Ireland, 3 Health Research Board, Dublin, Ireland, 4 Western Region Drug and Alcohol Task Force, Galway, Ireland, 5 Department of Public Health HSE South West, Cork, Ireland, 6 INFANT Research Centre, University College Cork, Cork, Ireland

* peter.barrett@ucc.ie

## Abstract

### Introduction

Stimulant drug use during adolescence (e.g., cocaine or ecstasy) can lead to a myriad of adverse health effects, but it remains uncertain how the home life environment and involvement in structured extracurricular activities may correlate with teenage stimulant use.

### Methods

We conducted an analysis utilising data from the 2020 Planet Youth survey, an anonymous questionnaire administered to school-going adolescents in the western region of Ireland. All adolescents with complete demographic information on age, gender and drug use were included in the analysis. Multivariable-adjusted logistic regression was used to explore factors associated with adolescent lifetime cocaine or ecstasy use. Exposure variables of interest were factors associated with the home environment (parental rule-setting, parental monitoring, being outside after midnight), mental health, intergenerational closure (defined as the extent of connectedness between adolescents and their peers, parents of their peers, and between parents themselves) and adolescent participation in extracurricular activities (sports, arts, volunteering, and/or afterschool clubs). Limitations of this study included its cross-sectional design which precluded causal inferences, self-reported data which may lead to information and social desirability biases, respectively, as well as adolescents who were not enrolled in formal education being excluded from the sampling frame.

which permits unrestricted use, distribution, and reproduction in any medium, provided the original author and source are credited.

**Data availability statement:** The data accessed for this study are part of the Planet Youth West of Ireland 2020 dataset, obtained through the Irish Social Science Data Archive (ISSDA) at University College Dublin, Ireland. Due to data protection policies and privacy restrictions, the authors are unable to share the underlying dataset. However, researchers may apply for access via ISSDA by completing the ISSDA Data Request Form for Research Purposes - Pseudonymised Datasets and submitting it to issda@ucd.ie. Further details on dataset access and usage policies can be found on the ISSDA website (https://www.ucd.ie/issda).

**Funding:** The author(s) received no specific funding for this work.

**Competing interests:** The authors have declared that no competing interests exist.

## Results

In total, 167 out of 4890 (3.4%) and 136 out of 4895 (2.8%) adolescents had previously used cocaine and ecstasy, respectively. Almost half of the participants in both datasets were female (49.7%), and the majority were aged 16 years (62.8% and 62.7% of the cocaine and ecstasy samples, respectively). Being outside after midnight during the previous week was associated with lifetime stimulant use (cocaine AOR = 5.63, 95%CI: 3.98,7.97; ecstasy AOR = 6.37, 95%CI: 4.36,9.30) (vs. never). Adolescents who reported "bad/very bad" mental health had over three times the odds of previous cocaine use (AOR = 3.63, 95%CI: 2.39, 5.49), and over five times the odds of previous ecstasy use (AOR = 5.15, 95%CI: 3.09, 8.59) (vs. "good/very good"). A lack of parental rule-setting (cocaine AOR = 1.28, 95%CI: 1.10, 1.50; ecstasy AOR = 1.39, 95%CI: 1.17,1.64), lack of parental monitoring (cocaine AOR = 1.81, 95%CI: 1.58,2.06; ecstasy AOR = 2.15, 95%CI: 1.86, 2.48), and reduced intergenerational closure (cocaine AOR = 1.37, 95%CI: 1.18,1.58; ecstasy AOR = 1.57, 95%CI: 1.34,1.83) were associated with lifetime stimulant use. Teenagers who did not participate regularly in sport had increased odds of previous cocaine use (in a club/team AOR = 1.50, 95%CI: 1.06,2.11; outside of a club/team AOR = 1.45, 95%CI: 1.03,2.04) and ecstasy use (in a club/team AOR = 1.54, 95%CI: 1.05,2.27; outside of a club/team AOR = 1.79, 95%CI: 1.56,3.12).

## Conclusion

The prevalence of stimulant use among this sample of Irish adolescents is relatively high by European standards, particularly cocaine use. Reduced parental rule-setting, supervision and family bonding time was associated with higher odds of adolescent stimulant use. Regular involvement in sports may have a protective effect against adolescent stimulant drug use.

---

## Introduction

The use of cocaine or ecstasy, commonly referred to as "stimulant drugs" [1,2], is common at a population level, and this is particularly true for cocaine [3,4]. In 2019, the Global Burden of Disease (GBD) study estimated that the global sum of disability-adjusted life years (DALYs) attributable to cocaine use disorder was 1.15 million, which represents a 114% increase from 1990 [5]. Notably, the DALYs attributable to cocaine use disorder begin to accelerate rapidly during adolescent years, primarily between the ages of 14–19 years [5].

Both cocaine and ecstasy carry a high risk of adverse health effects, including depression and suicidal ideation, seizures, myocardial infarctions, strokes, intense paranoia and/or hallucinations and lethal overdoses [6–10]. Moreover, during adolescence, the use of these substances can profoundly impact brain development, particularly in regions of the brain which undergo comparatively delayed maturation, such as the prefrontal cortex [11,12].

In 2019, Irish adolescents were burdened with the second highest prevalence of lifetime cocaine use (3.3%) in comparison to 34 other countries across Europe [13]. This estimate was approximately double the average European estimate (1.7%) and it slightly surpasses previous Irish estimates from 2003 and 2011, which were approximately 3% in both years [14,15]. Furthermore, the lifetime prevalence of ecstasy use among teenagers in the Republic of Ireland was 2.8% in 2019, which exceeded the overall average prevalence across Europe (2.3%) [13]. One possible explanation for higher prevalence estimates seen in Ireland may be a shift in perceived harms associated with stimulant drugs, which are increasingly seen as less dangerous or more socially acceptable, particularly among young people [16]. Moreover, cocaine is frequently used in social settings which also involve alcohol consumption, a pattern that may be especially relevant given that Ireland is an international outlier in terms of alcohol consumption [17,18]. These factors may partly explain why stimulant use trends in Ireland differ from those in other European nations. By contrast, during the past 20 years, Icelandic adolescents have reported a decline in both lifetime use of cocaine (3% in 2003 vs. 1.4% in 2019) and ecstasy (3% in 2003 vs. 1.3% in 2019), and this nation now boasts one of the lowest prevalence estimates of both drugs (as well as alcohol) across all European countries [13].

The notable reduction across a spectrum of substance misuse (i.e., alcohol, tobacco, cannabis, and stimulants) among Icelandic adolescents during the past two decades has largely been attributed to the Icelandic Prevention Model (IPM) [19]. The IPM is grounded in the Theory of Social Deviance [20], wherein adolescents are viewed as social products rather than fully autonomous decision-makers [21]. Consequently, substance use is viewed as a reflection of the broader social environment in which adolescents are embedded, rather than solely the result of individual choice. Thus, the IPM addresses distinct "risk" and protective factors across four primary community-based domains which overlap with one another. These include the home life environment, organised extracurricular activities, adolescents' peer groups, and the school setting [22]. One key concept within the home and community domains is intergenerational closure, which is the degree of connection, trust and communication between parents, their children, and the parents of their children's peers [23], and is considered a potential protective factor against adolescent substance use [24,25].

The decrease in adolescent substance use in Iceland has been accompanied by notable changes in home environment factors and greater adolescent participation in organised extracurricular activities. For example, between 2000–2016, there was a large decline in the proportion of adolescents who reported being outside of the home unsupervised after midnight at least once a week, decreasing from 80% to 31%, in addition to a rise in parental monitoring of their activities. The proportion of parents who were aware of whom their adolescent children were spending time with during the evenings rose from 50% to 74% [26]. Concurrently, regular adolescent engagement in organised sports with a club/team increased, and approximately 80% of all Icelandic youth now participate in organised sports at least once per week [27]. The apparent success of the IPM has led to its expansion on a global level, referred to as the "Planet Youth" project, which now involves collaboration between the Icelandic Centre for Social Research and Analysis (ICSRA) and other participating nations, including the Republic of Ireland [24]. Importantly, recent Planet Youth studies in Ireland have highlighted that similar home life environment factors and involvement in extracurricular activities may be associated with reduced likelihood of cannabis consumption [25], as well as binge drinking (albeit sports involvement may lead to increased levels of binge drinking among adolescents) [28].

To date, however, relatively few studies in Europe have explored the relationship between social and home life environment factors and adolescent stimulant drug use, and no previous study has examined these associations in Ireland. The IPM is particularly relevant to the study herein, as it forms the basis for the Planet Youth initiative and thereby provides a directly comparable framework through which to examine similar home and local level factors that may influence adolescent drug use. Therefore, this study aimed to investigate how specific home life factors and participation in organised extracurricular activities correlate with stimulant drug use among a sample of Irish adolescents using data from the Irish Planet Youth 2020 survey.

## Methods

### Study design

This cross-sectional study is a secondary analysis of the 2020 Planet Youth survey, entitled "Planet Youth - Growing Up in the West". This survey, conducted anonymously among secondary school-going students in the western region of Ireland (counties Galway, Mayo and Roscommon), was carried out in December 2020 and included 91 schools and educational settings (i.e., 100% of schools in the region). The questionnaire was adapted from the Icelandic survey developed by ICSRA and was translated into English. In total, the survey included 73 comprehensive questions, encompassing a broad spectrum of topics such as self-reported physical and mental well-being, substance use, family and peer relationships, school environment, internet and screen usage, leisure activities, sleep patterns and experiences of bullying [25].

### Study population and sampling

The study targeted all adolescent students within counties Galway, Mayo and Roscommon. Invitations were extended to 81 secondary schools and 10 Youthreach Centres (Department of Education and Skills programmes catering to early school leavers) [20]. Of the total 6321 students identified through school records, 5051 participated in the survey, resulting in a response rate of 80% [25]. The remaining students either withdrew (n = 30), or were unavailable on the designated survey day due to absenteeism or other commitments (n = 1240) [25]. The 5051 completed questionnaires underwent digitisation, data processing and validation in Reykjavik, Iceland, resulting in the exclusion of 47 additional surveys, yielding 5004 valid responses.

### Ethical considerations

The Planet Youth West research team obtained ethical approval for conducting these surveys through a passive consent process from the Royal College of Physicians of Ireland (RCPI). Passive consent meant that parents/guardians were informed about the study and given the opportunity to opt out on behalf of their child. Adolescents were also informed that participation was voluntary, and they could withdraw at any time. If parents/guardians wished to withdraw their child, they could do so by contacting the school before survey administration. The RCPI Ethics Committee approved this passive consent process, including the waiver of explicit written or verbal consent.

### Measurements

We utilised two separate datasets; one for cocaine use and one for ecstasy use. The anonymised datasets used in this study are securely stored on a password-protected computer and will be retained for at least 10 years after publication. Access to these data is strictly limited to the co-authors of this study.

### Dependent variables

Participants were asked the following questions: "How often have you used cocaine in your lifetime?" and "How often have you used ecstasy (E's, MDMA) in your lifetime?" We generated two binary outcome variables: previous (lifetime) cocaine use (yes/no) and previous (lifetime) ecstasy use (yes/no).

### Independent variables

Firstly, mental health was assessed using a single self-reported question: "How would you rate your mental health?" with response options that comprised of "very good", "good", "okay", "bad", or "very bad". These were collapsed into three categories: "Good/very good", "okay", and "bad/very bad".

Next, we examined four distinct variables related to the home environment. Initially, participants were asked how many times they were outside after midnight within the past week, and we subsequently collapsed categories to form a dichotomous variable; "were you outside after midnight in the previous week (yes/no)?"

Additionally, three scale variables were constructed for the remaining home environment factors, specifically parental rule-setting (e.g., prohibiting adolescent children from attending unsupervised parties), parental monitoring (e.g., regularly checking in via phone calls and ensuring a parent is present when an adolescent returns home), and intergenerational closure, i.e., the ongoing support, connections, and communication between different families across generations in a community, where both children and their parents are interconnected [23]. This latter construct captures the degree of integration and mutual oversight between families within a community which, as per the IPM, is believed to further increase monitoring and co-monitoring [24]. Parental rule-setting was assessed by participants' responses to the statements "my parents/carers set definite rules about what I can do at home" and "my parents/carers set definite rules about what I can do outside the home". Adolescents were asked how well these statements applied to them (very well; well; poorly; very poorly). These responses were originally scored from one (very well) to four (very poorly). However, we combined these responses (Cronbach's alpha = 0.789) to create a single variable (score range 2–8) and reverse coded them, whereby a lower score indicates decreased rule-setting.

Similarly, parental monitoring was assessed based on adolescents' responses to the statements "my parents/carers know who I am with in the evenings" and "my parents/carers know where I am in the evenings". Again, responses were combined (Cronbach's alpha = 0.847) and reverse coded, whereby lower scores indicate decreased parental monitoring. The same approach was taken for intergenerational closure, which was derived from responses to the statements "my parents/carers know my friends" and "my parents/carers know the parents of my friends" (Cronbach's alpha = 0.803). Higher scores represent stronger ties and increased likelihood of engagement between families, which may act as a protective factor against drug use.

Following the creation of these variables, scale standardisation was applied, wherein each standard deviation decrease on the scale corresponds to the odds ratio. This methodological approach aligns with previous research conducted by the Icelandic group [29], and other Irish studies investigating factors associated with adolescent substance misuse, specifically e-cigarette use and alcohol use using Planet Youth data [20,28].

The second set of independent variables pertained to respondents' engagement in structured extracurricular activities outside of school and also included four distinct variables. Specifically, adolescents were asked about their frequency of involvement in sports, arts (such as music, art, drama, or dance), volunteering, and/or other afterschool clubs (e.g., scouts, youth clubs, etc.). As the survey did not include specific measures of competitiveness or intensity, we tried to address this limitation by incorporating an additional variable that captures regular sports participation outside of a club or team (i.e., informal/casual play, although this may also include structured sports based on individual pursuit). This allowed us to partially assess the potential impact of less competitive sports engagement. Categories for each question were collapsed in order to synthesise binary variables (yes/no), indicating any participation or no participation in each activity.

In relation to potential confounders, this study incorporated six sociodemographic variables: age group, gender, relative income, family structure, maternal education and region of residence. Age group was defined as 15 years or less, 16 years or 17 years or more. Gender was categorised as male, female or other. Relative income was assessed by asking participants, "How well off financially do you think your family is in comparison to other families?", and responses were consolidated into three categories: better off, similar or worse off. Family structure was classified as either living with both parents or 'other'. Region of residence was divided into counties Galway, Mayo or Roscommon. Maternal education, serving as a proxy indicator of socioeconomic status, was determined by asking participants about the highest level of education completed by the mother or primary caregiver. This variable was then recoded into six categories: Masters/PhD, university degree, technical institute diploma, graduated secondary/high school, did not graduate secondary/high school,

or do not know/does not apply. All covariates included in our models were selected *a priori* based on a scoping review of international literature.

## Statistical analysis

The decision to analyse two separate datasets (one for cocaine use and one for ecstasy use) was made *a priori*, and was based on two primary considerations. First, given the relatively low prevalence of stimulant use in the sample (3.4% for cocaine and 2.8% for ecstasy), a combined multi-category outcome (e.g., "cocaine only", "ecstasy only", "both", "neither") would have resulted in small cell sizes for several subgroups, limiting statistical power and the reliability of estimates. Second, despite both being stimulant drugs, cocaine and ecstasy can have different patterns and contexts of use (e.g., cocaine is often used in conjunction with alcohol and may be more commonly integrated into routine social interactions (e.g., bars, parties) in comparison to ecstasy). This approach allowed for more targeted examination of correlates specific to each substance.

Descriptive characteristics were examined according to cocaine and ecstasy use. Categorical variables are shown as counts and percentages and continuous data are shown as a median and interquartile range. Cross-tabulation and Pearson's chi-square ($\chi^2$) tests were employed to identify differences in proportions and Mann Whitney U tests were used to compare parental rule-setting, parental monitoring and intergenerational closure scores between individuals with and without prior cocaine or ecstasy use. Binary logistic regression analyses were conducted to explore associations between each exposure variable and dichotomous outcomes adjusted for age group, gender, relative income, family structure, maternal education and region of residence. Data analyses were conducted using Stata version 17.1 (StataCorp., TX, USA) for Windows. For all analyses, a *p*-value (two tailed) of less than 0.05 was considered to indicate statistical significance.

## Results

### Descriptive characteristics

The final study samples were 4890 (97.7% of the study population) and 4895 (97.8% of the study population) in the cocaine and ecstasy datasets, respectively. Specifically, 114 and 109 participants, respectively, were removed due to missing data on age, gender, and/or cocaine/ecstasy use. Tables 1 and 2 present the characteristics of the study samples from the cocaine and ecstasy datasets, respectively. In total, 4723 (96.6%, 95% CI 96.0–97.1%) respondents had never used cocaine, whereas 167 (3.4%, 95% CI 2.9–4.0%) had previously used cocaine at some point in their lives. Furthermore, 4759 (97.2%, 95% CI 96.7–97.6%) participants had never used ecstasy, while 136 (2.8%, 95% CI 2.4–3.3%) had previously used ecstasy. Finally, 80 (1.6%, 95% CI 1.3–2.0%) had used both stimulants before.

With regard to demographic characteristics, 2429 participants (49.7%, 95% CI 48.3–51.1%) identified as female in the cocaine dataset and 2432 (49.7%, 95% CI 48.3–51.1%) identified as female in the ecstasy dataset. The majority of participants in both datasets were 16 years of age; 62.8% (95% CI 61.4–64.1%) and 62.7% (95% CI 61.3–64.0%) of respondents in the cocaine and ecstasy datasets, respectively. Among the full sample, most adolescents lived at home with both parents (79.5%, 95% CI 78.3–80.6%, and 79.4%, 95% CI 78.2–80.5% in the cocaine and ecstasy datasets, respectively). However, only 51.8% (95% CI 44.2–59.4%) and 49.6% (95% CI 41.2–58.1%) of adolescents who reported previous use of cocaine and ecstasy, respectively, lived at home with both parents. Among those who had used cocaine, 33.9% (95% CI 27.1–41.5%) reported being financially "better off" than others (vs. 43.8%, 95% CI 42.4–45.2% among those who never used cocaine), while the respective results for ecstasy were 38.1% (95% CI 30.2–46.6%) and 43.7% (95% CI 42.2–45.1%). With respect to extracurricular activities, more than one-half (52.8% in both datasets, 95% CI 51.4–54.2%) regularly participated in organised sports, almost three-quarters (73.5%; cocaine 95% CI 72.3–74.7%, ecstasy 95% CI 72.2–74.7%) regularly participated in sports outside of a club/team, while 28% (95% CI 26.8–29.3%) and 28.1% (95% CI

**Table 1. Descriptive characteristics among study participants in the cocaine dataset from the Planet Youth Ireland survey 2020.**

| Variables | Full sample n=4890 | Never used cocaine n=4763 | Used cocaine before n=167 | χ2 or ‡ | p-value |
|---|---|---|---|---|---|
| | n (%) | n (%) | n (%) | | |
| **Age group** | | | | 53.3 | **<0.001** |
| 15 years or less | 1331 (27.2) | 1301 (27.5) | 30 (18.0) | | |
| 16 years | 3071 (62.8) | 2978 (63.1) | 93 (55.7) | | |
| 17 years or more | 488 (10.0) | 444 (9.4) | 44 (26.3) | | |
| **Gender** | | | | 2.91 | 0.23 |
| Male | 2382 (48.7) | 2296 (48.6) | 86 (51.5) | | |
| Female | 2429 (49.7) | 2353 (49.8) | 76 (45.5) | | |
| Other | 79 (1.6) | 74 (1.6) | 5 (3.0) | | |
| **Relative income** | | | | 28.2 | **<0.001** |
| Better off | 2108 (43.5) | 2052 (43.8) | 56 (33.9) | | |
| Similar | 1983 (40.9) | 1924 (41.1) | 59 (35.8) | | |
| Worse off | 758 (15.6) | 708 (15.1) | 50 (30.3) | | |
| **Family structure** | | | | 80.5 | **<0.001** |
| Live with both parents | 3872 (79.5) | 3786 (80.4) | 86 (51.8) | | |
| Other | 1001 (20.5) | 921 (19.6) | 80 (48.2) | | |
| **Maternal education** | | | | 12.1 | **0.03** |
| Masters/PhD | 693 (14.3) | 675 (14.4) | 18 (11.0) | | |
| University degree | 1408 (29.1) | 1372 (29.3) | 36 (21.9) | | |
| Diploma from technical institute | 376 (7.8) | 357 (7.7) | 19 (11.6) | | |
| Graduated secondary school | 1094 (22.6) | 1054 (22.5) | 40 (24.4) | | |
| Did not graduate secondary school | 286 (5.9) | 270 (5.8) | 16 (9.8) | | |
| Do not know/Does not apply | 985 (20.3) | 950 (20.3) | 35 (21.3) | | |
| **Region** | | | | 2.1 | 0.35 |
| Galway | 2746 (56.2) | 2648 (56.1) | 98 (58.7) | | |
| Mayo | 1491 (30.5) | 1438 (30.4) | 53 (31.7) | | |
| Roscommon | 653 (13.3) | 637 (13.5) | 16 (9.6) | | |
| **Outside after midnight during week** | | | | 155.8 | **<0.001** |
| Yes | 655 (14.0) | 580 (12.8) | 75 (48.1) | | |
| **Mental health** | | | | 70.8 | **<0.001** |
| Good/very good | 2278 (46.8) | 2236 (47.5) | 42 (25.3) | | |
| Okay | 1590 (32.6) | 1542 (32.8) | 48 (28.9) | | |
| Bad/very bad | 1002 (20.6) | 926 (19.7) | 76 (45.8) | | |
| **Parental rule-setting** | | | | 3.79 | **<0.001** |
| Median (IQR)* | 6.0 (1.0) | 6.0 (1.0) | 5.0 (2.0) | | |
| **Intergenerational closure^** | | | | 5.83 | **<0.001** |
| Median (IQR)* | 7.0 (2.0) | 7.0 (2.0) | 6.0 (2.0) | | |
| **Parental monitoring** | | | | 8.32 | **<0.001** |
| Median (IQR)* | 7.0 (2.0) | 7.0 (2.0) | 6.0 (2.0) | | |
| **Regular participation in a sports club/team** | | | | 17.1 | **<0.001** |
| No | 2290 (47.2) | 2185 (46.6) | 105 (62.9) | | |
| **Regular participation in sports outside of a club/team** | | | | 12.9 | **<0.001** |
| No | 1284 (26.5) | 1220 (26.0) | 64 (38.6) | | |
| **Regular participation in music/art/drama/dance** | | | | 3.4 | 0.06 |
| No | 3503 (72.0) | 3373 (71.7) | 130 (78.3) | | |

*(Continued)*

**Table 1.** (Continued)

| Variables | Full sample n=4890 | Never used cocaine n=4763 | Used cocaine before n=167 | χ2 or ‡ | p-value |
|---|---|---|---|---|---|
| | n (%) | n (%) | n (%) | | |
| **Regular participation in volunteering** | | | | 1.1 | 0.29 |
| No | 3720 (76.7) | 3587 (76.6) | 133 (80.1) | | |
| **Regular participation in afterschool clubs** | | | | 1.6 | 0.20 |
| No | 3938 (81.1) | 3797 (81.0) | 141 (84.9) | | |

IQR, Interquartile range; Significant *p*-values shown in bold.

‡Refers to the Mann-Whitney U test statistic.

*Lower median scores correspond to lower parental rule-setting, monitoring, and intergenerational closure scores.

^Intergenerational closure is defined as the extent of connectedness between adolescents and their peers, parents of their peers, and between parents themselves.

26.9–29.4%) regularly participated in music/art/drama/dance, respectively. Finally, 23.3% (cocaine 95% CI 22.1–24.5%, ecstasy 95% CI 22.2–24.5%) of adolescents in both datasets regularly participated in volunteering activities.

## Multivariable-adjusted logistic regression

The results from binary logistic regression models are shown in Table 3 (cocaine) and Table 4 (ecstasy).

Self-reported mental health was strongly associated with stimulant use in adjusted models. Adolescents who described their mental health as "bad/very bad" had over threefold increased odds of previous cocaine use (AOR=3.63, 95%CI: 2.39, 5.49) and over fivefold increased odds of previous ecstasy use (AOR=5.15, 95%CI: 3.09, 8.59) (vs. those reporting "good/very good" mental health). Those who described their mental health as "okay" had elevated odds of ecstasy use (AOR=2.64, 95%CI: 1.58, 4.42).

Regarding the home life environment, in fully adjusted models, adolescents who were outside after midnight during the previous week were over five times more likely to have previously used cocaine (AOR=5.63, 95% CI: 3.98, 7.97) and had a six-fold increased odds of previous ecstasy use (AOR=6.37, 95% CI: 4.36, 9.30) (vs. never). Each one standard deviation decrease in the rule-setting score was associated with a 28% increased odds of previous cocaine use (AOR=1.28, 95% CI: 1.10, 1.50) and a 39% increased odds of previous ecstasy use (AOR=1.39, 95% CI: 1.17, 1.64) (vs. never). Furthermore, every one standard deviation decrease in the intergenerational closure score was significantly associated with an increased odds of previous cocaine use (AOR=1.37, 95% CI: 1.18, 1.58) and previous ecstasy use (AOR=1.57, 95% CI: 1.34, 1.83) (vs. never). Each one standard deviation decrease in the parental monitoring score was associated with an approximately 80% increased odds of cocaine use (AOR=1.81, 95% CI: 1.58, 2.06), and a 115% increased odds of ecstasy use (AOR=2.15, 95% CI: 1.86, 2.48) (vs. never).

With respect to participation in extracurricular activities, lack of regular adolescent involvement in sports outside the school (whether in a club/team or not) was the only extracurricular activity that was significantly associated with lifetime cocaine and/or ecstasy use in fully adjusted models. Specifically, adolescents who did not regularly participate in sports in a club/team were 50% more likely to have engaged in previous cocaine use (AOR=1.50, 95% CI: 1.06, 2.11), and had a 54% increased odds of previous ecstasy use (AOR=1.54, 95% CI: 1.05, 2.27) (vs. never). Adolescents who did not regularly participate in sports outside of a club/team were 45% more likely to have engaged in previous cocaine use (AOR=1.45, 95% CI: 1.03, 2.04), and were 79% more likely to have engaged in previous ecstasy use (AOR=1.79, 95% CI: 1.24, 2.59) (vs. never).

A sensitivity analysis of the groups with and without any missing information (i.e., those with at least one exposure and/or confounding variable missing) did not yield a statistically significant association between the presence of missing data and use of cocaine ($\chi^2$=2.07; p=0.15) or ecstasy ($\chi^2$=0.72; p=0.40).

**Table 2. Descriptive characteristics among study participants in the ecstasy dataset from the Planet Youth Ireland survey 2020.**

| Variables | Full sample n=4895 | Never used ecstasy n=4759 | Used ecstasy before n=136 | χ2 or ‡ | p-value |
|---|---|---|---|---|---|
| | n (%) | n (%) | n (%) | | |
| **Age group** | | | | 64.6 | <0.001 |
| 15 years or less | 1335 (27.3) | 1313 (27.6) | 22 (16.2) | | |
| 16 years | 3069 (62.7) | 2996 (62.9) | 73 (53.7) | | |
| 17 years or more | 491 (10.0) | 450 (9.5) | 41 (30.1) | | |
| **Gender** | | | | 2.2 | 0.34 |
| Male | 2383 (48.7) | 2313 (48.6) | 70 (51.5) | | |
| Female | 2432 (49.7) | 2370 (49.8) | 62 (45.6) | | |
| Other | 80 (1.6) | 76 (1.6) | 4 (2.9) | | |
| **Relative income** | | | | 21.3 | <0.001 |
| Better off | 2111 (43.5) | 2060 (43.7) | 51 (38.1) | | |
| Similar | 1983 (40.9) | 1940 (41.1) | 43 (32.1) | | |
| Worse off | 759 (15.6) | 719 (15.2) | 40 (29.8) | | |
| **Family structure** | | | | 75.1 | <0.001 |
| Live with both parents | 3872 (79.4) | 3805 (80.2) | 67 (49.6) | | |
| Other | 1006 (20.6) | 938 (19.8) | 68 (50.4) | | |
| **Maternal education** | | | | 11.0 | 0.05 |
| Masters/PhD | 691 (14.2) | 672 (14.3) | 19 (14.2) | | |
| University degree | 1411 (29.1) | 1384 (29.4) | 27 (20.1) | | |
| Diploma from technical institute | 376 (7.8) | 366 (7.8) | 10 (7.5) | | |
| Graduated secondary school | 1095 (22.6) | 1063 (22.5) | 32 (23.9) | | |
| Did not graduate secondary school | 286 (5.9) | 271 (5.7) | 15 (11.2) | | |
| Do not know/Does not apply | 988 (20.4) | 957 (20.3) | 31 (23.1) | | |
| **Region** | | | | 0.09 | 0.96 |
| Galway | 2749 (56.2) | 2672 (56.1) | 77 (56.6) | | |
| Mayo | 1490 (30.4) | 1450 (30.5) | 40 (29.4) | | |
| Roscommon | 656 (13.4) | 637 (13.4) | 19 (14.0) | | |
| **Outside after midnight during week** | | | | 154.8 | <0.001 |
| Yes | 654 (14.0) | 588 (12.9) | 66 (51.6) | | |
| **Mental health** | | | | 68.9 | <0.001 |
| Good/very good | 2279 (46.8) | 2255 (47.6) | 24 (17.9) | | |
| Okay | 1590 (32.6) | 1542 (32.5) | 48 (35.8) | | |
| Bad/very bad | 1005 (20.6) | 943 (19.9) | 62 (46.3) | | |
| **Parental rule-setting** | | | | 4.59 | <0.001 |
| Median (IQR)* | 6.0 (1.0) | 6.0 (1.0) | 5.0 (2.0) | | |
| **Intergenerational closure^** | | | | 7.29 | <0.001 |
| Median (IQR)* | 7.0 (2.0) | 7.0 (2.0) | 6.0 (2.0) | | |
| **Parental monitoring** | | | | 9.39 | <0.001 |
| Median (IQR)* | 7.0 (2.0) | 7.0 (2.0) | 6.0 (3.0) | | |
| **Regular participation in a sports club/team** | | | | 17.3 | <0.001 |
| No | 2290 (47.2) | 2202 (46.7) | 88 (64.7) | | |
| **Regular participation in sports outside of a club/team** | | | | 21.1 | <0.001 |
| No | 1285 (26.5) | 1226 (26.0) | 59 (43.7) | | |
| **Regular participation in music/art/drama/dance** | | | | 5.6 | 0.02 |
| No | 3503 (71.9) | 3393 (71.6) | 110 (80.9) | | |
| **Regular participation in volunteering** | | | | 0.3 | 0.58 |
| No | 3720 (76.7) | 3613 (76.6) | 107 (78.7) | | |

*(Continued)*

**Table 2.** (Continued)

| Variables | Full sample n=4895 | Never used ecstasy n=4759 | Used ecstasy before n=136 | χ2 or ‡ | p-value |
|---|---|---|---|---|---|
| | n (%) | n (%) | n (%) | | |
| **Regular participation in afterschool clubs** | | | | 2.9 | 0.09 |
| No | 3938 (81.1) | 3820 (80.9) | 118 (86.8) | | |

IQR, Interquartile range; Significant *p*-values shown in bold.

‡Refers to the Mann-Whitney U test statistic.

*Lower median scores correspond to lower parental rule-setting, monitoring, and intergenerational closure scores.

^Intergenerational closure is defined as the extent of connectedness between adolescents and their peers, parents of their peers, and between parents themselves.

## Discussion

To our knowledge, this is the first study to examine both home life environment and extracurricular activity variable associations with adolescent stimulant drug use, both in Ireland and internationally. The lifetime prevalence of cocaine use among our study sample was 3.4%, while the lifetime prevalence of ecstasy use was 2.8%. These figures are consistent with those observed in the European School Survey Project on Alcohol and Other Drugs (ESPAD) report from 2019 (3.3% for lifetime cocaine use and 2.8% for lifetime ecstasy use), which was a nationally representative study [13]. Overall, approximately 1.6% of adolescents had used both stimulants before. Of note, despite the predominantly rural population in western Ireland, people living in this region exhibit many similar sociodemographic characteristics in comparison to the broader Irish population, and thus, these findings are likely to be generalisable to adolescents elsewhere in Ireland, as well as internationally [30–32]. Given that adolescence is a critical juncture for neurodevelopment, these findings underscore the need for comprehensive evidence-based strategies to curtail stimulant use. This is particularly true for cocaine, given that Irish adolescents are burdened with the second highest lifetime prevalence in Europe [13].

In this study, it was found that adolescents who were socialising outside after midnight at least once during the previous week were significantly more likely to have used both cocaine and ecstasy. Venturing outdoors unsupervised late at night is a well-recognised factor which can increase the probability of substance misuse among adolescents, as adolescents are more likely to encounter situations where drugs are being used, most notably by their own peers [24,29]. Our findings also demonstrate that a lack of parental rule-setting was associated with lifetime use of cocaine and ecstasy. However, prior research on parental rule-setting and its association with teenage drug use has been somewhat conflicting. While some research has found that parental rule-setting can reduce the likelihood of teenage drug use (similar to the study herein) [33,34], other evidence has suggested that stringent rule-setting may be associated with an increased propensity for adolescent drug use [35]. Chaplin and colleagues found that *open discussions* about drug use situations can create a more comfortable environment for adolescents, and may therefore be associated with a reduced likelihood of use [35]. It is plausible that establishing boundaries, whilst also allowing for open dialogue about drug use, may cultivate a mutual trust between parents and their adolescent children, potentially reducing the likelihood of teenage substance misuse, whereas the strict imposition of extensive rules might relate to a feeling of curtailed independence, thereby leading to rebellious behaviour.

We also observed that cocaine and ecstasy use were significantly more likely among teenagers who reported lower levels of parental monitoring, which is consistent with previous evidence in relation to both stimulant drugs [36–40]. However, in the Irish context, no study has examined this association to date, hence our study provides novel insights into the Irish setting specifically. Our research additionally noted that participants who reported reduced levels of intergenerational

**Table 3. Binary logistic regression of cocaine use among study participants in the Planet Youth Ireland survey 2020.**

| Variables | Previously used vs. never | 95% CI | Previously used vs. never | 95% CI |
|---|---|---|---|---|
| | Crude OR | | Adjusted OR‡ | |
| **Outside after midnight during week** | | | | |
| No | 1 | $p < 0.001$ | 1 | $p < 0.001$ |
| Yes | **6.30** | Ref (4.55, 8.73) | **5.63** | Ref (3.98, 7.97) |
| **Mental health** | | | | |
| Good/very good | 1 | $p = 0.02$ | 1 | $p < 0.001$ |
| Okay | **1.66** | Ref | 1.40 | Ref (0.93, 2.23) |
| Bad/very bad | **4.37** | (1.09, 2.52) (2.97, 6.42) | 3.63 | (2.39, 5.49) |
| **Parental rule-setting** | | | | |
| 1 SD decrease corresponds to | **1.41** | $p < 0.001$ (1.22, 1.64) | **1.28** | $p = 0.002$ (1.10, 1.50) |
| **Intergenerational closure^** | | | | |
| 1 SD decrease corresponds to | **1.51** | $p < 0.001$ (1.32, 1.73) | **1.37** | $p < 0.001$ (1.18, 1.58) |
| **Parental monitoring** 1 SD decrease corresponds to | **1.88** | $p < 0.001$ (1.67, 2.13) | **1.81** | $p < 0.001$ (1.58, 2.06) |
| **Regular participation in a sports club/team** | | | | |
| Yes | 1 | $p < 0.001$ | 1 | $p = 0.02$ |
| No | **1.94** | Ref (1.41, 2.67) | **1.50** | Ref (1.06, 2.11) |
| **Regular participation in sports outside of a club/team** | | | | |
| Yes | 1 | $p < 0.001$ | 1 | $p = 0.03$ |
| No | **1.78** | Ref (1.29, 2.45) | **1.45** | Ref (1.03, 2.04) |
| **Regular participation in music/art/drama/dance** | | | | |
| Yes | 1 | $p = 0.07$ | 1 | $p = 0.29$ |
| No | 1.42 | Ref (0.98, 2.07) | 1.24 | Ref (0.83, 1.85) |
| **Regular participation in volunteering** | | | | |
| Yes | 1 | $p = 0.29$ | 1 | $p = 0.64$ |
| No | 1.23 | Ref (0.84, 1.82) | 1.10 | Ref (0.73, 1.65) |
| **Regular participation in afterschool clubs** | | | | |
| Yes | 1 | $p = 0.20$ | 1 | $p = 0.29$ |
| No | 1.32 | Ref (0.86, 2.04) | 1.28 | Ref (0.81, 2.02) |

Ref, Reference category; SD, Standard deviation; CI, Confidence interval; Significant $p$-values shown in bold.

‡Adjusted for age group, gender, relative income, family structure, maternal education, and region of residence.

^Intergenerational closure is defined as the extent of connectedness between adolescents and their peers, parents of their peers, and between parents themselves.

closure were more inclined to report cocaine and ecstasy use. To our knowledge, this is the first study to quantify the association between intergenerational closure and adolescent stimulant drug use.

Our research found a strong association between poorer self-reported mental health and stimulant drug use. This finding aligns with previous evidence [7–9], suggesting a need for family and school- and community-based mental health

**Table 4. Binary logistic regression of ecstasy use among study participants in the Planet Youth Ireland survey 2020.**

| Variables | Previously used vs. never Crude OR | 95% CI | Previously used vs. never Adjusted OR‡ | 95% CI |
|---|---|---|---|---|
| **Outside after midnight during week** | | | | |
| No | 1 | $p < 0.001$ | 1 | $p < 0.001$ |
| Yes | **7.18** | Ref (5.02, 10.26) | 6.37 | Ref (4.36, 9.30) |
| **Mental health** | | | | |
| Good/very good | 1 | $p < 0.001$ | 1 | $p < 0.001$ |
| Okay | **2.92** | Ref | 2.64 | Ref |
| Bad/very bad | **6.18** | (1.78, 4.79) (3.83, 9.96) | 5.15 | (1.58, 4.42) (3.09, 8.59) |
| **Parental rule-setting** | | | | |
| 1 SD decrease corresponds to | **1.55** | $p < 0.001$ (1.32, 1.82) | 1.39 | $p < 0.001$ (1.17, 1.64) |
| **Intergenerational closure^** | | | | |
| 1 SD decrease corresponds to | **1.74** | $p < 0.001$ (1.50, 2.01) | 1.57 | $p < 0.001$ (1.34, 1.83) |
| **Parental monitoring** | | | | |
| 1 SD decrease corresponds to | **2.20** | $p < 0.001$ (1.92, 2.51) | 2.15 | $p < 0.001$ (1.86, 2.48) |
| **Regular participation in a sports club/team** | | | | |
| Yes | 1 | $p < 0.001$ | 1 | $p = 0.03$ |
| No | **2.10** | Ref (1.47, 2.99) | 1.54 | Ref (1.05, 2.27) |
| **Regular participation in sports outside of a club/team** | | | | |
| Yes | 1 | $p < 0.001$ | 1 | $p = 0.002$ |
| No | **2.21** | Ref (1.56, 3.12) | 1.79 | Ref (1.24, 2.59) |
| **Regular participation in music/art/drama/dance** | | | | |
| Yes | 1 | $p = 0.02$ | 1 | $p = 0.06$ |
| No | **1.68** | Ref (1.09, 2.58) | 1.57 | Ref (0.99, 2.48) |
| **Regular participation in volunteering** | | | | |
| Yes | 1 | $p = 0.58$ | 1 | $p = 0.98$ |
| No | 1.13 | Ref (0.74, 1.71) | 1.00 | Ref (0.65, 1.56) |
| **Regular participation in afterschool clubs** | | | | |
| Yes | 1 | $p = 0.09$ | 1 | $p = 0.21$ |
| No | 1.54 | Ref (0.93, 2.55) | 1.40 | Ref (0.83, 2.36) |

Ref, Reference category.

SD, Standard deviation.

CI, Confidence interval.

‡Adjusted for age group, gender, relative income, family structure, maternal education, and region of residence.

Significant *p*-values shown in bold.

^Intergenerational closure is defined as the extent of connectedness between adolescents and their peers, parents of their peers, and between parents themselves.

supports as part of comprehensive prevention strategies. Although our study could not determine the direction of causality due to its cross-sectional design, the strong associations observed suggest that interventions targeting adolescent mental health could potentially reduce the risk of stimulant drug use.

In our study sample, regular engagement in sports was protective against teenage cocaine and ecstasy use, which is consistent with previous evidence [41]. Interestingly, the impact of regular sports engagement on adolescent substance use has been a topic of some debate. While the majority of published evidence has shown that it appears to be protective against stimulant drug use (as well as so-called softer drugs, such as cannabis), active engagement in sports has previously been associated with other forms of teenage substance use, such as initiation of alcohol use [24,41,42]. In this study, other extracurricular activities (music/art/drama/dance, volunteering, and/or afterschool clubs) did not yield a significant association with adolescent stimulant use. However, participation levels were considerably lower for these activities when compared to sports, e.g., 28% for music/art/drama/dance and 23% for volunteering (compared to 53% for sports in a club/team, or 74% for sports outside of a club/team). As such, reduced statistical power may have limited our ability to detect true associations for these variables, given that only a minority of adolescents ever used stimulants.

Notwithstanding potential data limitations, in 2007, the municipal government of Reykjavík, Iceland, introduced a prepaid voucher initiative known as the "Recreation Card" to enhance adolescent participation in a broad array of extracurricular activities [43]. As of January 1st, 2023, this annual voucher is valued at 75,000 Icelandic króna (approximately €515 based on current exchange rates), and is made available to all Icelandic residents aged 6–18 years [43]. This programme forms a key part of the IPM and has been linked to high engagement in sports, with about 80% of Icelandic adolescents taking part in organised sports clubs at least weekly (in comparison to approximately 50% of adolescents in our study) [27]. Notably, a similar national "Recreation Card" scheme, with a starting annual value of €130 per person, has already been proposed in Ireland to increase adolescent access to a range of extracurricular activities across the country [25,44]. However, this policy has not yet been funded or implemented.

In addition to a national Recreation Card scheme, a schools-based primary prevention strategy that may curtail teenage substance use is the implementation of a mandatory "Social, Personal and Health Education (SPHE)" subject at both primary and secondary level. Since 2024, SPHE has been made a mandatory subject across Irish schools [45], and it provides age-appropriate instruction about life skills relating to emotional well-being, responsible decision-making (including decisions surrounding drug use) and digital awareness. This universal intervention may be particularly relevant given the rise of social media platforms and encrypted messaging services that facilitate anonymous drug trading [46,47].

As previously discussed, Irish adolescents have the second highest prevalence of cocaine use across Europe [13]. In addition, while some progress was made in curtailing the prevalence of adolescent ecstasy use in Ireland between 2000 and 2010, this previous decline has plateaued over the past decade [13–15]. The exact reasons which underpin this are likely multifaceted and complex but may include the advent of new social media platforms as well as the increasing popularity of the dark web which has resulted in new opportunities for illicit drug trading [46], and potential targeting of adolescents. For example, Snapchat includes features such as temporary photos and self-erasing messages which may provide a sense of anonymity and security for both buyers and sellers of illicit substances, as it generates minimal digital footprints, thereby mitigating the risk for both parties involved [46]. In relation to the latter, lockdowns and restrictions due to the COVID-19 pandemic caused substantial disruptions to traditional drug supply chains worldwide, leading to a marked increase in the use of the dark web [47]. For example, between January 2020 and March 2020 the European Monitoring Centre for Drugs and Drug Addiction (EMCDDA), now the European Union Drugs Agency (EUDA), reported a notable rise (approximately 25%) in online drug trading, primarily for five main substances: cannabis, cocaine, ecstasy, sedatives and opioids [48]. In the study herein, 37 (22%) and 41 (30%) adolescents who had used cocaine and ecstasy, respectively, had previously purchased their illicit drugs online. Unfortunately, we cannot draw comparisons between this finding from the 2020 survey and those of the initial Planet Youth results from 2018 (i.e., in a pre-pandemic era), as questions pertaining to online drug trading were newly included in the 2020 questionnaire. Nevertheless, the ongoing inclusion

of these questions in future Planet Youth surveys is important, as they could potentially highlight changes in online drug purchasing among adolescents in Ireland, which may complicate public health efforts to curtail substance misuse among this demographic.

**Strengths and limitations**

There are a number of strengths to this study. First, this is the only Irish study to date to examine the home life environment and organised extracurricular activity variable associations with adolescent stimulant drug use. Second, 91 schools (i.e., 100% of schools in the western region of Ireland) who were formally extended invitations agreed to participate, and the final response rate was high (80%) [25]. Generally, a response rate of 80% or higher is considered exemplary [49], and this reduces non-response bias [50]. Third, the anonymity of responses may have further enhanced the internal validity of our study due to the curtailment of social desirability bias, particularly regarding topics of a sensitive nature, e.g., drug use. Fourth, we had relatively complete data (<1% missing data for each variable). Finally, the size of our study samples afforded adequate statistical power to detect true relationships.

However, a number of limitations should be considered. First, given the cross-sectional nature of the study, elucidating the precise chronological order of events is not possible, thereby preventing us from establishing any causal links. Thus, longitudinal studies are required to assess how changes in the home life environment and extracurricular involvement over time influence the use of stimulant drugs. Second, all data were self-reported, raising the possibility of information bias, such as inconsistent answers due to participant inattention, or how adolescents' reporting of parental monitoring and rule-setting may not capture parents' perceptions of these socially constructed factors. In order to reduce the former, future iterations of the survey in Ireland will be conducted digitally via computers, flagging inconsistencies in real-time in order to validate responses. For the latter, although logistically more complex, future surveys could consider triangulating adolescent data on some specific responses (e.g., relating to parental monitoring or supervision) with complementary reports from parents/guardians. Third, although the survey was anonymous and self-completed, the influence of social desirability bias cannot be ruled out, e.g., adolescents may have overreported positive behaviours, such as participation in extracurricular activities. Fourth, although stratification by type of institution (secondary school vs. Youthreach centre) was considered *a priori*, the dataset herein does not allow us to decipher whether participants attended secondary schools or Youthreach centres. This is an important limitation, as data from a national survey among youths in Ireland suggest that adolescents who attend Youthreach centres are more likely to come from socioeconomically disadvantaged backgrounds, and may be at higher risk of substance misuse [51]. Indeed, stimulant drug use can be socially patterned, and bivariate analysis herein found a significant association between socioeconomic status and stimulant use. Fifth, some adolescents in the region may not be engaged in formal education and were therefore not included in the sampling frame. Although this number is expected to be low, unfortunately we do not have access to these figures. Finally, given that the Planet Youth 2020 surveys took place in December 2020 (after a period of national lockdown due to the COVID-19 pandemic), it will be important to repeat these surveys at future timepoints to monitor trends in stimulant drug use, as well as trends in extracurricular activity participation.

**Conclusion**

This study indicates that the use of stimulant drugs is relatively high among adolescents in Ireland, particularly cocaine. Socialising outside after midnight in the previous week, poorer self-reported mental health, reduced parental monitoring, parental rule-setting, intergenerational closure, and lack of regular participation in sports were all associated with adolescent use of both cocaine and ecstasy. These findings may provide valuable insights for actionable prevention interventions at the community level, as well as for policy-makers and educators. These may include adjustments to curricular content, expansion of extra-curricular activities offered to adolescents, and the introduction of a national Recreation Card scheme for young people.

## Acknowledgments

We express our appreciation to the Western Region Drug and Alcohol Task Force (WRDATF) for their efforts in distributing surveys to each school and for their active role in disseminating the reports, in addition to the Icelandic Centre for Social Research and Analysis (ICSRA) for their work in cleaning, processing, and returning the data used to generate the dataset for this study. Additionally, we extend our gratitude to all the adolescents who took part in the survey.

## Author contributions

**Conceptualization:** Fionn P. Daly, Peter M. Barrett.

**Formal analysis:** Fionn P. Daly.

**Methodology:** Fionn P. Daly, Seán R. Millar, Peter M. Barrett.

**Supervision:** Seán R. Millar, Peter M. Barrett.

**Writing – original draft:** Fionn P. Daly.

**Writing – review & editing:** Fionn P. Daly, Seán R. Millar, Emmet Major, Peter M. Barrett.

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
