## [Decision Letter · Decision Letter 0]

28 Mar 2025

PONE-D-25-05204Prevention of adolescent stimulant drug use: What are the roles of home life environment and extracurricular activities? Findings from the Irish Planet Youth Survey 2020PLOS ONE?

Dear Dr. Barrett,

Thank you for submitting your manuscript to PLOS ONE. After careful consideration, we feel that it has merit but does not fully meet PLOS ONE’s publication criteria as it currently stands. Therefore, we invite you to submit a revised version of the manuscript that addresses the points raised during the review process.

We look forward to receiving your revised manuscript.

Kind regards,

Karen M Davison, PhD

Academic Editor

PLOS ONE

Additional Editor Comments (if provided):

Reviewers' comments:

Reviewer's Responses to Questions

**Comments to the Author**

1. Is the manuscript technically sound, and do the data support the conclusions?

Reviewer #1: Yes

Reviewer #2: Yes

2. Has the statistical analysis been performed appropriately and rigorously?

Reviewer #1: Yes

Reviewer #2: Yes

3. Have the authors made all data underlying the findings in their manuscript fully available?

Reviewer #1: Yes

Reviewer #2: Yes

4. Is the manuscript presented in an intelligible fashion and written in standard English?

Reviewer #1: Yes

Reviewer #2: Yes

Reviewer #1: This manuscript effectively highlights important findings on adolescent stimulant use in Ireland and its associations with parental rule-setting, monitoring, intergenerational closure, and extracurricular activities participation. With some clarifications, streamlined explanations, and stronger policy recommendations, it could have a greater impact on public health discourse.

KEY STRENGTHS

Robust Dataset: The study draws from a large sample size (n=~5000) with a high response rate (80%), enhancing generalizability.

Novel Insights: It provides new evidence on intergenerational closure as a potential protective factor against adolescent stimulant use.

Policy Relevance: Findings can inform family-based and extracurricular interventions in Ireland, especially considering the high cocaine use prevalence among Irish adolescents.

STRENGTHEN JUSTIFICATION FOR STUDY

The study compares Irish data with Icelandic trends. It may be useful to briefly mention why Iceland’s model is particularly relevant.

Highlight why stimulant use trends in Ireland may differ from those in other European nations.

RESULTS

Data Interpretation:

Some findings (e.g., sports participation being protective) are expected, but the lack of significant association for volunteering and arts deserves further explanation.

Table Presentation:

Consider highlighting statistically significant findings in bold for clarity.

Ensure consistency in decimal places across all tables.

METHODS

Define Variables More Clearly:

The operationalization of intergenerational closure is unclear. A more precise definition or detailed description and explanation of its measurement would be beneficial.

Parental rule-setting and monitoring could be elaborated with specific examples.

Extracurricular Activities:

Since participation is self-reported, consider discussing the potential for social desirability bias.

The study only examined frequency of participation. Would depth (e.g., competitive vs. Recreational/casual sports) influence the results?

LIMITATIONS

Self-Report Bias: While acknowledged, this could be expanded with suggestions for mitigating this bias in future studies.

Causal Limitations: Emphasize that cross-sectional studies do not establish causation and suggest longitudinal approaches for future research.

Potential Sample Bias: It is stated that 100% of schools in the region participated, but does this sample reflect national trends?

CONCLUSION

The conclusion restates findings but could provide a stronger call to action regarding policy implementation.

Consider adding specific recommendations for intervention strategies rather than just emphasizing the need for further research.

Reviewer #2: Authors report on the study examining the cross-sectional relationship between indicators of home life and community activities among youth and use of cocaine and/ or ecstasy among high school students in Western Ireland. In a sample of approximately 4900 high school students, the current study identifies that approximately 3.4% of Irish high school youth have used cocaine and 2.8% have used ecstasy. Associated risk factors include a lack of parental rule setting, a lack of parental monitoring, and reduced inter- generational closure defined as how well parents know the peers their children spend time with and the peers’ parents. In addition, students who did not participate regularly in sports were at greater risk of previous cocaine and or ecstasy use.

The study uses a survey adapted from research that informed a successful intervention program developed in Iceland which led to diminished use of illicit substances among youth in the early 2000s.

Concerns:

1. The underlying theoretical model or rationale for choice of indicators regarding home life, community activities, etc. should be more clearly specified in the introduction.

2. The rationale for presenting the data as two separate but parallel studies is not clear. Presumably the same sample were used for all the data. Can the study be presented as a single data set with two outcomes in this way authors could also present information regarding the youth who used both kinds of illicit stimulants.

3. The details of the sample (age/grade, proportion female) are not presented clearly in the abstract or in the initial description of the sample.

4. Did the demographic question clearly inquire about gender or sex at birth? Was non-binary an option?

5. The stated study objective was to estimate prevalence of cocaine and ecstasy use among Irish adolescents using data from the Irish planet youth 2020 survey. However, it does not appear that estimating a population prevalence was what the study was designed to accomplish. Rather authors reported data from a community sample of youth who attended schools and who answered the survey questions. The study objective needs to be more focussed on the goals of examining associations between specific home life factors, organized part extracurricular activities, and the use of illicit stimulants.

6. How closely linked is the study with the research program in Iceland?

7. How many Irish youth do not attend school in these counties? Who is missing from the sampling frame?

8. What response options were available for the questions measuring lifetime cocaine use or ecstasy?

9. The term “intergenerational closure” is new to this reader. The definition could be included in the introduction.

10. I suggest referencing/ including previous Irish studies of substance misuse using the same sample in the introduction. Were there similar findings with home life variables?

11. Please clarify the information that the planet youth research offers in distinction to that provided by the EMCDDA.

**Do you want your identity to be public for this peer review?** For information about this choice, including consent withdrawal, please see our Privacy Policy

Reviewer #1: **Yes: ** Edore Onigu-Otite

Reviewer #2: No

---

## [Author Response · Author response to Decision Letter 1]

14 May 2025

Please find detailed Response to Reviewers in accompanying letter of response.

---

## [Decision Letter · Decision Letter 1]

9 Jun 2025

PONE-D-25-05204R1Prevention of adolescent stimulant drug use: What are the roles of home life environment and extracurricular activities? Findings from the Irish Planet Youth SurveyPLOS ONE?

Dear Dr. Barrett,

Thank you for submitting your manuscript to PLOS ONE. After careful consideration, we feel that it has merit but does not fully meet PLOS ONE’s publication criteria as it currently stands. Therefore, we invite you to submit a revised version of the manuscript that addresses the points raised during the review process.

We look forward to receiving your revised manuscript.

Kind regards,

Karen M Davison, PhD

Academic Editor

PLOS ONE

Journal Requirements:

Additional Editor Comments:

Please align definitions of intergenerational closure provided in the abstract and footnoted in your results tables

Acknowledge limitations of your study in the abstract

Please confirm if mental health diagnosis e.g., depression, anxiety was available in the survey. If it was please include in analysis. If not, please highlight that this was an important covariate you were unable to control for in your limitations

Reviewers' comments:

Reviewer's Responses to Questions

**Comments to the Author**

Reviewer #2: (No Response)

2. Is the manuscript technically sound, and do the data support the conclusions?

Reviewer #2: Yes

3. Has the statistical analysis been performed appropriately and rigorously?

Reviewer #2: Yes

4. Have the authors made all data underlying the findings in their manuscript fully available?

Reviewer #2: Yes

5. Is the manuscript presented in an intelligible fashion and written in standard English?

Reviewer #2: Yes

Reviewer #2: Thank you for your responses,

And for the clear explanation as to why the study was completed as two separate analyses. I suggest including a brief version of this rationale in the statistical analysis section of manuscript.

In the results section, description of the rates of cocaine and ecstasy use (as well as other variables) are offered as percentages. Please include Confidence intervals for these estimates of percentage.

Suggest that Tables 1 and 2 can be simplified by offering only a single value for the yes/ no binary variables, eg., can present only yes, as the no is understood.

The statement regarding combined cocaine and ecstasy should appear in the results rather than in discussion for the first time.

In the first paragraph of the discussion, do you have a comment about other countries that have looked at this question regarding home life environment and extracurricular activities and stimulant drug use (in addition to Iceland)

**Do you want your identity to be public for this peer review?** For information about this choice, including consent withdrawal, please see our Privacy Policy

Reviewer #2: No

---

## [Author Response · Author response to Decision Letter 2]

1 Jul 2025

Please see attached Response Letter to Reviewers

---

## [Decision Letter · Decision Letter 2]

4 Aug 2025

Prevention of adolescent stimulant drug use: What are the roles of home life environment and extracurricular activities? Findings from the Irish Planet Youth Survey

PONE-D-25-05204R2

Dear Dr. Dr. Barrett:

We’re pleased to inform you that your manuscript has been judged scientifically suitable for publication and will be formally accepted for publication once it meets all outstanding technical requirements.

Kind regards,

Karen M Davison, PhD

Academic Editor

PLOS ONE

Additional Editor Comments (optional):

Reviewers' comments:

Reviewer's Responses to Questions

**Comments to the Author**

Reviewer #2: All comments have been addressed

2. Is the manuscript technically sound, and do the data support the conclusions?

Reviewer #2: Yes

3. Has the statistical analysis been performed appropriately and rigorously?

Reviewer #2: Yes

4. Have the authors made all data underlying the findings in their manuscript fully available?

Reviewer #2: Yes

5. Is the manuscript presented in an intelligible fashion and written in standard English?

Reviewer #2: Yes

Reviewer #2: Thank you for the opportunity to evaluate this manuscript. All my concerns have now been addressed.

**Do you want your identity to be public for this peer review?** For information about this choice, including consent withdrawal, please see our Privacy Policy

Reviewer #2: No

---

## [Editor Report · Acceptance letter]

PONE-D-25-05204R2

PLOS ONE

Dear Dr. Barrett,

I'm pleased to inform you that your manuscript has been deemed suitable for publication in PLOS ONE. Congratulations! Your manuscript is now being handed over to our production team.

Kind regards,

on behalf of

Dr. Karen M Davison

Academic Editor

PLOS ONE